# Comprehensive Comparative Analysis and Development of Molecular Markers for *Dianthus* Species Based on Complete Chloroplast Genome Sequences

**DOI:** 10.3390/ijms232012567

**Published:** 2022-10-19

**Authors:** Shengnan Lin, Jianyi Liu, Xingqun He, Jie Wang, Zehao Wang, Xiaoni Zhang, Manzhu Bao, Xiaopeng Fu

**Affiliations:** 1Key Laboratory of Horticultural Plant Biology, College of Horticulture and Forestry Sciences, Huazhong Agriculture University, Wuhan 430070, China; 2Guangdong Laboratory for Lingnan Modern Agriculture, Genome Analysis Laboratory of the Ministry of Agriculture, Agricultural Genomics Institute at Shenzhen, Chinese Academy of Agricultural Sciences, Shenzhen 518120, China

**Keywords:** *Dianthus*, chloroplast genome, phylogenic analysis, molecular markers

## Abstract

*Dianthus* spp. is a genus with high economic and ornamental value in the Caryophyllaceae, which include the famous fresh-cut carnation and the traditional Chinese herbal medicine, *D. superbus*. Despite the *Dianthus* species being seen everywhere in our daily lives, its genome information and phylogenetic relationships remain elusive. Thus, we performed the assembly and annotation of chloroplast genomes for 12 individuals from seven *Dianthus* species. On this basis, we carried out the first comprehensive and systematic analysis of the chloroplast genome sequence characteristics and the phylogenetic evolution of *Dianthus*. The chloroplast genome of 12 *Dianthus* individuals ranged from 149,192 bp to 149,800 bp, containing 124 to 126 functional genes. Sequence repetition analysis showed the number of simple sequence repeats (SSRs) ranged from 75 to 80, tandem repeats ranged from 23 to 41, and pair-dispersed repeats ranged from 28 to 43. Next, we calculated the synonymous nucleotide substitution rates (Ks) of all 76 protein coding genes to obtain the evolution rate of these coding genes in *Dianthus* species; *rpl22* showed the highest Ks (0.0471), which suggested that it evolved the swiftest. By reconstructing the phylogenetic relationships within *Dianthus* and other species of Caryophyllales, 16 *Dianthus* individuals (12 individuals reported in this study and four individuals downloaded from NCBI) were divided into two strongly supported sister clades (Clade A and Clade B). The Clade A contained five species, namely *D. caryophyllus*, *D. barbatus*, *D. gratianopolitanus*, and two cultivars (‘HY’ and ‘WC’). The Clade B included four species, in which *D. superbus* was a sister branch with *D. chinensis*, *D. longicalyx*, and F_1_ ‘87M’ (the hybrid offspring F_1_ from *D. chinensis* and ‘HY’). Further, based on sequence divergence analysis and hypervariable region analysis, we selected several regions that had more divergent sequences, to develop DNA markers. Additionally, we found that one DNA marker can be used to differentiate Clade A and Clade B in *Dianthus*. Taken together, our results provide useful information for our understanding of *Dianthus* classification and chloroplast genome evolution.

## 1. Introduction

*Dianthus* is an important genus of Caryophyllaceae, comprising approximately 600 species, which are distributed in Europe, Asia, and Africa; the main producing area is the Mediterranean region [1]. Most of them are perennial herbs and rare annuals [2]. The name *Dianthus* comes from the Greek words “dios”, meaning divine, and “anthos”, meaning flower. The genus *Dianthus* contains four sections, which are Barbulatum, Carthusianum, Dianthus, and Fimbriatum [3]. For the first section, Barbulatum, these include *D. chinensis*, *D. elatus*, *D. ramosissimus*, *D. repens*, and *D. turkestanicus*. Among them, *D. chinensis* is usually used as a ground cover, in flower beds, or as a potted plant [4]. For the second section, Carthusianum, *D. barbatus* is often used as fresh cut flowers because of its large inflorescence. For the third section, Dianthus, *D. caryophyllus* is the most famous for providing high-quality cut flowers, commonly known as mothers’ flowers. The fourth section is Fimbriatum; the petals are deeply notched, giving them a feathery or fringed appearance, and the flower has a rich and charming fragrance. Additionally, *D. superbus* is a kind of traditional Chinese medicine, where the whole plant can be used as medicine, and its effect is to clear away heat, diuresis, break blood, and clear meridians. Furthermore, it can also be used as a pesticide to kill insects [5,6]. Therefore, *Dianthus* plants not only have extremely high ornamental and economic value, but they also have great medicinal properties. However, many cultivated varieties of the *Dianthus* species on the market are obtained through multiple hybridization, and the genetic background is unclear [7]. So, it is difficult to distinguish between them using morphological and histological authentication methods. Additionally, the genetic divergence among these species and the complex evolutionary history of the genus are often poorly understood, making it difficult for the bioprospecting of the *Dianthus* species.

In recent years, with the rapid development of sequencing technology and the reduction in sequencing costs, the chloroplast genome sequence is easier to obtain and has a lower cost than the nuclear genome. At the same time, the chloroplast genome is small, and the structural and coding genes are relatively conservative, which makes the chloroplast genome an important part of molecular evidence for the study of higher plant phylogeny [8]. At present, a variety of plants have used the chloroplast genome to study the phylogenetic relationship, such as *Spondias*, *Epimedium*, Olive, and Asteraceae [9,10,11,12]. Chloroplast genome-based phylogenetic relationships provide new insights and reflections on our understanding of plant evolution. Moreover, the conserved features of land plant chloroplast genomes could be used as the ideal markers for super-barcoding to separate indiscernible species groups. One example is that the DNA barcodes (*matK*, *rbcL*, *matK-rps16*, *ycf1,* and *ycf3*) from the complete chloroplast genome of the *Sanguisorba* species could distinguish the typical *S. officinalis* and *S. officinalis* var. longifolia [13]. However, there are no systematic studies to develop DNA markers in *Dianthus* based on the chloroplast genome.

The first plastome of the *Dianthus* species (*D. gratianopolitanus*) was released in 2015. Later, three other chloroplast genomes from *D. caryophyllus*, *D. longicalyx*, and *D. chinensis* were published or released [1,14]. A few phylogenetic studies have been conducted on *Dianthus*, but the interspecific relationships in this genus remain controversial. In addition, the existing data are not sufficient to comprehensively illustrate the intricate phylogenetic relationships within the *Dianthus* genus. A robust phylogeny of *Dianthus*, including more representative species and a large amount of genetic markers, is essential for understanding the evolutionary history, the breeding of new cultivars, and the conservation of *Dianthus* germplasm resources. Here, we assembled the complete chloroplast genomes of 12 individuals from seven *Dianthus* species, followed by their comparison with four previously reported *Dianthus* chloroplast genomes from NCBI. This study aimed to: (1) contribute new fully sequenced chloroplast genomes in *Dianthus* and improve the understanding of the overall structure of these genomes; (2) perform a comparative analysis of the chloroplast genomes of these 16 *Dianthus* individuals; (3) reconstruct the robust phylogenetic relationship of *Dianthus* using chloroplast genome evidence, allowing for an examination of their concordance with current taxonomy; and (4) develop novel DNA markers to discriminate *Dianthus* species. The results obtained in this study can improve our understanding of the classification, phylogeny, and evolution of this important ornamental and medicinal genus.

## 2. Results

### 2.1. Features and Characteristics of the Dianthus Complete Chloroplast Genomes

Complete chloroplast genome lengths for the 12 individuals from seven *Dianthus* species (six *D. chinensis*: Dch ‘MH’, Dch ‘dhs’, Dch ‘cf’, Dch ‘L’, Dch ‘X’, and Dch ‘DPD’; one *D. superbus*: Dsu ‘QM’; one *D. barbatus*: Dbr ‘XB’; one *D. caryophyllus*: Dca ‘XSZH’; two cultivars: ‘HY’, ‘WC’; one hybrid: F_1_ ‘87M’) ranged from 149,192 bp to 149,800 bp (Figure 1; Table 1). All the chloroplast genomes showed a typical quadripartite structure comprising a large single-copy (LSC) region (82,436–82,963 bp) and a small single-copy (SSC) region (17,096–17,227 bp) separated by two inverted repeat (IR) regions (24,781–24,818 bp) (Figure 2 and Appendix A; Table 1). The average GC content was ~36.31% (Table 1). Among the 12 *Dianthus* chloroplast genomes, Dbr ‘XB’ and Dca ‘XSZH’ contained 124 functional genes; Dch ‘MH’, Dsu ‘QM’, ‘HY’, and ‘WC’ contained 126 functional genes; and the others contained 125 genes. For the protein coding genes, Dch ‘MH’, Dsu ‘QM’, Dbr ‘XB’, Dca ‘XSZH’, ‘HY’, and ‘WC’ had 84 protein coding genes; the others had 83 (Table 2; Figure 3a). With regard to transfer RNA (tRNA), only Dbr ‘XB’ had 33 tRNA genes, the rest had 34. As for ribosomal RNA (rRNA), all of the other 11 *Dianthus* chloroplast genomes encoded a set of eight rRNA genes, except Dca ‘XSZH’, which did not have *rrn23S*. Sixteen protein coding genes had two copies (*ndhB*, *rpl2*, *rps7*, *rps12*, *ycf1*, *ycf2*, *rrn16S*, *rrn23S*, *rrn4.5S*, *rrn5S*, *trnA-UGC*, *trnI-GAU*, *trnL-CAA*, *trnN-GUU*, *trnR-ACG*, and *trnV-GAC*) (Table 2). There were 15 genes containing one intron (*rps16*, *atpF*, *rpoC1*, *ycf3*, *petB*, *petD*, *rpl16*, *ndhA*, *ndhB*, *trnA-UGC*, *trnI-GAU*, *trnK-UUU*, *trnL-UAA*, *trnV-UAC*, and *trnI-GAU*), while three genes, *clpP*, *ycf3* and *rps12*, possessed two introns in all of the 12 *Dianthus* chloroplast genomes (Table 2 and Appendix A). In particular, *rps12* is generated via trans-splicing (Table 2 and Appendix A). Interestingly, *ycf1* had one intron in Dca ‘XSZH’ and ‘HY’, and *rps19* had one intron in Dch ‘cf’ (Table 2 and Appendix A).

### 2.2. Sequence Repetition in the Dianthus Complete Chloroplast Genomes

A comparative analysis of sequence repetition between all 12 chloroplast genomes found that the overall distribution, types, and numbers of repeats are highly similar among the *Dianthus* species. The number of SSRs identified in the 12 chloroplast genomes ranged from 75 to 80 (Table 3 and Appendix A). The most abundant mononucleotide SSRs are polyadenine or polythymine repeat types (Table 3). It is interesting to note that tetranucleotide SSRs were not found in Dbr ‘XB’, Dca ‘XSZH’, ‘WC’, and ‘HY’, and hexanucleotide SSRs were only detected in Dch ‘cf’, Dsu ‘QM’, Dbr ‘XB’, ‘WC’, and ‘HY’ (Table 3). Furthermore, there were two types of pentanucleotide SSRs, the repeat units being AACAC/GTGTT and AATAC/ATTGT. Except for Dch ‘MH’ and F_1_ ‘87M’, 10 other chloroplast genomes had two AACAC/GTGTT repeat units (Table 3). Differently, the AATAC/ATTGT repeat unit was not detected in Dch ‘cf’, and the others possessed one copy (Table 3). SSRs were more frequently located in the LSC regions (44–48 loci) compared to the SSC regions (13–14 loci) and IR regions (6–8 loci) of the 12 sequenced chloroplast genomes (Figure 3b; Appendix A). Most of the SSRs were identified in the intergenic regions (36–40 loci), followed by the protein coding regions (17–21 loci) and introns (10–13 loci) (Figure 3c; Appendix A). These SSRs could be used to develop potential molecular markers for species differentiation and population genetics in future research.

Tandem repeats analysis of *Dianthus* chloroplast genomes showed that the numbers of tandem repeats ranged from 23 in Dch ‘cf’, up to 41 in Dbr ‘XB’ and ‘WC’. The copy numbers of these repeats ranged from 2 to 26.3 copies per tandem repeat, and the repeat sizes ranged from 3 bp to 105 bp per copy (Appendix A). The tandem repeats were found extensively in the intergenic regions, and most were located in the LSC regions (Appendix A). In addition, there were 28–43 pairs of dispersed repeats, which belonged to forward, reverse, complementary, and palindromic repeats in 12 *Dianthus* chloroplast genomes (Appendix A). Forward (direct) and palindrome (inverted) repeats were considerably higher in number than reverse and complement repeats. Most notably, complement repeats were not detected in the chloroplast genome sequences of Dsu ‘QM’, Dbr ‘XB’, Dca ‘XSZH’, ‘WC’, and ‘HY’ (Figure 3d; Appendix A). The majority of these repeats, with the repeat length ranging from 30 bp to 39 bp, were located in intergenic regions. We found that the dispersed repeats within these genes were mostly located in the LSC regions (Figure 3e; Appendix A). They can potentially facilitate structural rearrangements and develop variability among plastomes in a population (Appendix A).

### 2.3. Contraction and Expansion of IR

The variations in the single-copy and IR regions’ sizes and boundaries commonly cause evolutionary events, such as contraction and expansion in the plastome architecture [16]. A comprehensive comparison at the LSC/IR/SSC boundaries was performed among the 16 individuals from nine *Dianthus* species, including the 12 individuals sequenced in our research and four that were downloaded from NCBI. Although the structural boundaries of the 16 individual chloroplast genomes were highly conserved, structural variations were still found in the LSC/IR/SSC boundary regions (Figure 4). From an overall viewpoint, the chloroplast genome length of *D. caryophyllus* (Dca, downloaded from NCBI) is the shortest, although it has the longest LSC regions when compared with other species. Additionally, using Dca as a reference, 15 individuals’ IR regions were expanded significantly. The *rps19* gene was located in the LSC/IRb regions, except for Dca and Dca ‘XSZH’. Particularly, the *ycf2* gene spanned the LSC and IRb regions only in Dca species. In addition, the *ndhF* gene did not cross the IRb region, which was only located in the SSC region in ‘HY’ species.

### 2.4. Evolutionary Rates Analyses

We calculated Ks of all 76 protein coding genes to obtain the evolution rates of these coding genes in *Dianthus* species. Among plastid genes within *Dianthus*, *rpl22* showed the highest Ks (0.0471), which suggested that they evolved the swiftest, followed by *rps4* (0.0409), *ndhA* (0.0279), *rps18* (0.0278), and *rpl32* (0.0273) (Figure 5a; Appendix A). In contrast, nearly 40 genes, such as *ycf3*, *rps2*, *rps16*, *rps15*, *rps14*, *rps12*, etc., were the slowest evolving plastid genes, with a Ks of 0 or extremely close to 0, indicating their high degree of conservation.

In order to explore the evolutionary rates of different regions, we divided these genes into groups according to distribution. It can be seen that although the LSC region had the most genes, the Ks is relatively conservative, while the SSC region showed the highest evolution rate. Additionally, the IR regions had the fewest genes and showed comparatively high conservation. These indicate that compared with the LSC and IR regions, the SSC region is the most prone to mutation (Figure 5b). From the functional classification of genes, unknown genes, which include four *ycf* genes, showed the highest Ks, followed by other genes, self-replication genes and photosynthesis genes (Figure 5c). The results are highly consistent with function. As we know, photosynthesis is the most basic function of chloroplasts, and maintaining the stability of photosynthesis can maintain the normal growth of plants.

### 2.5. Phylogenetic Analyses

To study the phylogenetic position of 16 individuals from nine *Dianthus* species within the *Dianthus* genus, we performed a phylogenetic analysis using the maximum likelihood (ML) method and Bayesian Inference (BI) analyses based on six matrixes. The reconstructed phylogeny indicated that Caryophyllales was paraphyletic and that the species from Caryophyllaceae (*Gymnocarpos*, *Spergula*, *Colobanthus*, *Pseudostellaria*, *Agrostemma*, *Silene*, *Psammosilene*, *Gypsophila*, and *Dianthus*) were deemed non-monophyletic (Appendix A). *Dianthus* and *Gypsophila* diverged together from a common ancestor. As for the phylogenetic relationship within *Dianthus*, we obtained a total of 12 topologies according to six matrixes with two methods (Figure 6 and Appendix A). The 12 topological structures represented six possible phylogenetic relationships of *Dianthus*, and the six possible phylogenetic relationships, all supporting 16 individuals from nine *Dianthus* species, can be clustered into two major clades. Further, T1 (3/12) and T4 (3/12) are the main topological structures that occupy the same proportion. The difference between T1 and T4 was the location of the Dgr species. In T1, Dgr was sister to ‘HY’, and ‘WC’ and Dbr ‘XB’, while Dgr was sister to the subclade that contains Dsu ‘QM’, Dlo, and *D. chinensis* (Dch, Dch ‘MH’, Dch ‘cf’, Dch ‘dhs’, Dch ‘X’, Dch ‘L’, Dch ‘DPD’, and F_1_ ‘87M’) in T4. In general, for the phylogenetic position of Dgr, most of the 12 topological relationships (8/12) supported that Dgr belonged to the branch with ‘HY’, and ‘WC’ and Dbr ‘XB’. Therefore, we believe that T1 was more representative of the phylogenetic relationship of these 16 individuals from nine *Dianthus* species.

As T1 showed, two strongly supported clades (Clade A and Clade B) were recognizable. Clade A contained two *D. caryophyllus* (Dca and Dca ‘XSZH’), one *D. barbatus* (Dbr ‘XB’), one *D. gratianopolitanus* (Dgr), and two cultivars (‘HY’ and ‘WC’); Clade B contained one *D. superbus* (Dsu ‘QM’), one hybrid (F_1_ ‘87M’), one *D. longicalyx*, and six *D. chinensis* (Dch ‘MH’, Dch ‘dhs’, Dch ‘cf’, Dch ‘L’, Dch ‘X’, and Dch ‘DPD’). Both Clade A and Clade B were further diversified into two subclades; they highly corresponded to sect. Dianthus, sect. Carthusianum, sect. Fimbriatum, and sect. Barbulatum, as classified by Flora of China [3]. This phylogenetic relationship implied that sect. Dianthus and sect. Carthusianum shared the closest phylogenetic relationship, and sect. Fimbriatum and sect. Barbulatum showed a close relationship. It was worth noting that ‘HY’ and ‘WC’ were grouped into sect. Carthusianum, indicating that their female parent was native to sect. Carthusianum. Similarly, the female parent of F_1_ ‘87M’ was *D. chinensis*, and so it was sister to Dch ‘MH’.

**Figure 6 ijms-23-12567-f006:**
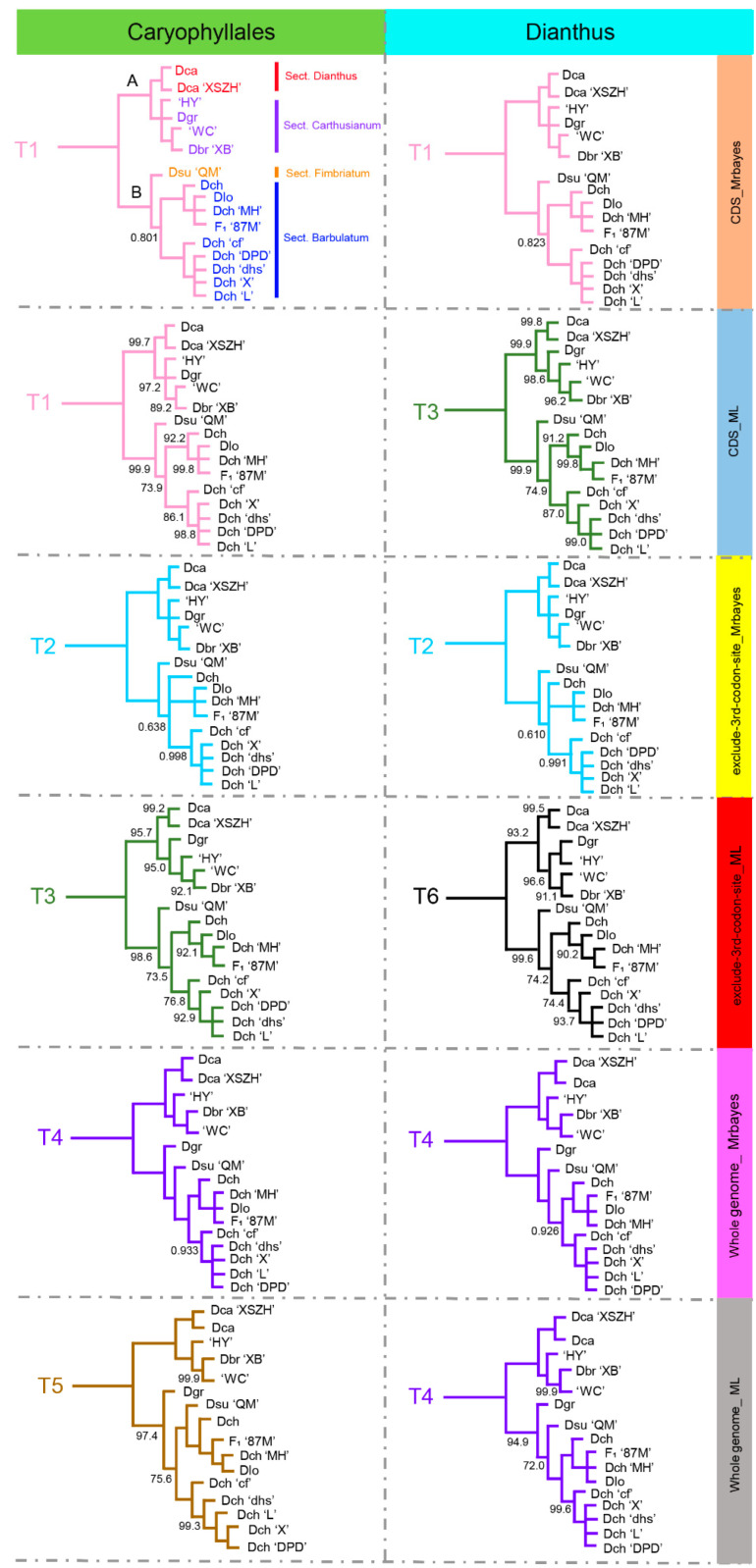
Twelve topologies for 16 *Dianthus* species based on six datasets from Caryophyllales and *Dianthus* chloroplast genomes. The six topologies on the left used 46 Caryophyllales species. The six topologies on the right used 16 *Dianthus* species and two outgroups. The label on the right indicates the type of dataset used and the method of constructing the tree. The same color represents the same topology. For all trees, unlabeled nodes have 100% support.

### 2.6. Hypervariable Regions

The results of the comprehensive sequence divergence of the 12 newly assembled individuals, with Dca ‘XSZH’ as the control displaying high sequence similarity (Appendix A). As expected, the IR regions were less divergent than the LSC and SSC regions. The coding regions were more conserved than the noncoding regions. However, the *psaA*, *ycf3*, *clpP*, *psbT*, *psbN*, *rpl16*, *rps19*, *ycf1*, and *ndhA* genes, and the *trnT-GGU*, *trnT-UGU*, *trnL-UAA*, *trnV-UAC*, *trnM-CAU*, and *trnL-UAG* RNAs, showed a relatively high degree of sequence divergence. Additionally, the intergenic spacer regions were highly diverse, particularly in the following regions: *rps16-trnR-UCU*, *aptF-aptI*, *rpoB-psbD*, *rps4-trnL-UAA*, *trnP-UGG-psbB*, *trnS-UGA-rps14*, *rpl16-rpl2*, and *rpl32-ycf1*.

Next, the nucleotide diversity (Pi) values within 500 bp windows were calculated, to detect the sequence divergence hot spots among the *Dianthus* chloroplast genomes (Figure 7). The Pi values were in the range of 0–0.01283, with two regions having peaks; these were *atpB* and *ycf1*, separately. Divergence hot spot regions could be the ideal molecular markers to distinguish *Dianthus* species.

### 2.7. Molecular Marker Development Based on Dianthus Plastomes

To discriminate the 12 individuals from seven *Dianthus* species, we selected several hypervariable regions to develop DNA markers, and only one marker could successfully distinguish some *Dianthus* species. Here, we only show results with valid markers. There was one valid DNA marker in the *clpP-psbB* (a part of *trnP-UGG-psbB* region) region. The one marker was used to differentiate *D. caryophyllus* (Dca ‘XSZH’), *D. barbatus* (Dbr ‘XB’), and two cultivars (‘WC’ and ‘HY’) from *D. superbus* (Dsu ‘QM’), *D. chinensis* (Dch ‘MH’, Dch ‘DPD’, Dch ‘X’, Dch ‘L’, Dch ‘dhs’, and Dch ‘cf’), and hybrid offspring F_1_ (F_1_ ‘87M’) (Figure 8). According to the results of phylogenetic analysis, *D. caryophyllus* (Dca ‘XSZH’), *D. barbatus* (Dbr ‘XB’), and two cultivars (‘WC’ and ‘HY’) were grouped into Clade A, and *D. superbus* (Dsu ‘QM’), *D. chinensis* (Dch ‘MH’, Dch ‘DPD’, Dch ‘X’, Dch ‘L’, Dch ‘dhs’, and Dch ‘cf’), and hybrid offspring F_1_ (F_1_ ‘87M’) were grouped into Clade B. The results of the DNA markers were consistent with the results of the phylogenetic analysis, which verifies the correctness and rationality of the results of the phylogenetic analysis.

## 3. Discussion

In this study, we reported 12 complete chloroplast genomes of the *Dianthus* species. Through the assembly and annotation of these genomes, we obtained more detailed information on the chloroplast genome of *Dianthus*, and we present a comparative analysis.

There was some degree of variation in the chloroplast genome lengths of 12 individuals from seven *Dianthus* species, with ‘HY’ having the longest genome size, which was 149,800 bp, and F_1_ ‘87M’ being the shortest, at only 149,192 bp (Table 1). The sizes of the *Dianthus* plastomes in this study were almost identical to the other four reported plastomes, which were 149,570 bp (*D. chinensis*, NC_053731), 149,665 bp (*D. gratianopolitanus*, LN877392), 147,604 bp (*D. caryophyllus*, NC_039650), and 149,596 bp (*D. longicalyx*, NC_050834) [1,14]. Repeated sequences can mediate rearrangements in the genome, provide resources for genome variation, and promote the evolution of species [17]. By analyzing the numbers, types, and distributions of the repeat sequences in all 12 chloroplast genomes, it was found that their patterns are basically the same (Table 3 and Appendix A). Simultaneously, most repeat sequences are extensively in the intergenic regions, and most are located in the LSC regions (Figure 3c; Appendix A). Similar conditions are found in other species, such as *Salvia* species, Zingiberaceae species, and *Dalbergia* species [8,18,19]. The coding regions contained most of the conserved sequences, whereas the non-coding regions had most of the variable sequences, indicating that the evolution rates of different regions of the plant chloroplast genome are significantly different. The functional gene coding region sequence is restricted by natural selection pressure, and the evolution rate is relatively slow, while the evolution rate of the unrestricted non-coding region sequence is faster [18,20,21].

Contraction and expansion at the borders of the IR regions of chloroplast genomes are considered to be important evolutionary events, and they may cause size variations, the origination of pseudogenes, gene duplication, or the reduction of duplicate genes to single copies [22,23,24]. Judged from a comparative analysis with the plastome of Dca as a reference, the IR lengths for all of the 12 individuals in *Dianthus* species plastomes sequenced in this study were increased to approximately 24,800 bp (Figure 4). The strongest evidence supporting this expansion is that the entire *ycf2* gene of other *Dianthus* species is located in the IRb region, while the *ycf2* gene of Dca species has a part (246 bp) that is located in the LSC region (Figure 4). Additionally, the *rps19* gene changed from being located only in the LSC region in the Dca species, to spanning both the LSC and IRb regions in other species, apart from Dca ‘XSZH’. The length of the *rps19* gene in Dca ‘XSZH’ was 150 bp, which was only approximately half the length of the normal *rps19* gene (Figure 4).

Nucleotide variations that do not result in amino acid changes are called synonymous mutations. It is generally believed that synonymous mutations are not subject to natural selection, and thus Ks can reflect the background base substitution rate of the evolutionary process [16,25]. Therefore, in order to understand the evolutionary history of *Dianthus* species, we calculated the Ks of all 76 protein coding genes in 16 *Dianthus* individuals, with most genes showing high conservation. (Figure 5a). Among them, the *rpl22* gene evolved the swiftest and had the highest Ks (0.0471). Further, most of the slower evolutionary genes were related to photosynthesis and self-replication (Figure 5c). Due to their essential functions, these genes are very conservative, and they do not tend to change, similar to the characteristics shown in many other plants [26,27,28]. Meanwhile, we observed the overall situation of Ks in the LSC, SSC, and IR regions, which showed that the SSC region had higher Ks values than the two others (Figure 5b). To some extent, this situation shows that the overall evolutionary rate of coding genes located in the SSC region is higher than that in the LSC and IR regions. Based on the analysis results of the repetitive sequences, the conservation of the LSC and SSC regions will be lower than that of the IR regions. Consistent with the results of similar studies in other plants, the LSC and SSC regions were less conserved than the IR regions [8,29].

As we all know, *Dianthus* was classified into four sections on the basis of the morphological features, such as inflorescence morphology, petal characteristics, and capsule shape, by Flora of China [3]. In the past, molecular markers, such as SSR [30], RAPD (randomly amplified polymorphic DNA) [31], SRAP (Sequence-related amplified polymorphism), and ISSR (inter-simple sequence repeat) [32], were used to distinguish the cultivars of *Dianthus* species. However, the genetic backgrounds of the cultivars were unclear, and these markers were not informative for inferring the relationships of those species. The chloroplast genome has become an efficient option for increasing plant phylogenomics at multiple taxonomic levels over the past few years [33,34,35,36,37]. For instance, the phylogenies analyzed using the complete chloroplast genomes of three *Spondias* species revealed a robust phylogenetic topology for *Spondias* [9]. A phylogenetic analysis of 32 species in the family Asteraceae demonstrated the phylogenetic position of the woody Sonchus alliance within the tribe Cichorieae and the sister relationship between the weedy Sonchus oleraceus and the alliance [12]. Similarly, based on 45 plastomes from 32 species of *Epimedium*, the molecular phylogeny, the infrageneric classification, the divergence times, and the ancestral states for flower traits were analyzed. These findings provide new insights into the relationships among *Epimedium* species and pave the way for a better elucidation of the classification and evolution of this genus [10]. Furthermore, robust phylogenetic relationships for *Lagerstroemia* species were reconstructed using different plastome sequence partitions and multiple phylogenetic methods for the first time [38]. Additionally, 11 *Olive europaea* were divided into two main groups, and *O. europaea* subsp. cuspidata formed a separate group (Cuspidata group) with the other subspecies (Mediterranean/North African group) from the reconstruction of phylogenetic relationships through the chloroplast genomes of 11 *O. europaea* [11]. These studies suggest that the chloroplast genome facilitates the analysis of the phylogenetic relationships of hybrid-bred modern cultivars. Thus, we used chloroplast genome data to infer the phylogenetic relationships of 16 *Dianthus* individuals, and we discovered that the chloroplast genome sequences had effective information for inferring the phylogeny of this genus.

Here, we recovered a well-supported and species-level relationship of Caryophyllales and *Dianthus* using six different chloroplast genome datasets through two methods (Figure 6 and Appendix A). Our phylogenetic analysis results create reliable phylogenies of the *Dianthus* species sequenced, using chloroplast genome information for the first time. Congruent with previous studies that used the nuclear ribosomal internal transcribed spacer (ITS) region, and five chloroplast genes and intergenic spacers (*matK*, *ndhF*, *trnL-trnF*, *trnQ-rps16*, and *trnS-trnfM*), Caryophyllales was paraphyletic, and the species from Caryophyllaceae, including *Gymnocarpos*, *Spergula*, *Colobanthus*, *Pseudostellaria*, *Agrostemma*, *Silene*, *Psammosilene*, *Gypsophila,* and *Dianthus* were deemed non-monophyletic (Appendix A) [39]. In addition, *Dianthus* and *Gypsophila* diverged together from a common ancestor (Appendix A).

Of course, our analyses strongly identified that 16 individuals from nine *Dianthus* species can be clustered into two major clades and further subdivided into four sections, namely sect. Dianthus, sect. Carthusianum, sect. Fimbriatum, and sect. Barbulatum. Additionally, our classification results based on the chloroplast genome were highly consistent with the classification results of morphological features. Simultaneously, this provided strong support for sect. Dianthus and sect. Carthusianum sharing the closest phylogenetic relationship, and sect. Fimbriatum and sect. Barbulatum showing a close relationship. The positions of the two cultivars and one hybrid in the phylogeny should be noted. For F_1_ ‘87M’, it was the hybrid offspring F_1_ from *D. chinensis* (♀) and ‘HY’ (♂), and it was clustered into sect. Barbulatum. This result confirmed that the chloroplast of *Dianthus* was of matrilinear inheritance. Therefore, according to phylogenetic position, we inferred the female parent of the two cultivars, namely, ‘HY’ and ‘WC’, from sect. Carthusianum. In addition, according to the taxonomy from Flora of China, Dlo belonged to the sect. Fimbriatum and should be clustered with Dsu ‘QM’, but our phylogenetic relationship showed that it clustered with sect. Barbulatum. This contradictory result may suggest that Dlo was a hybrid with its female parent from sect. Barbulatum. Moreover, according to the inferred phylogeny from ML and BI analyses using different datasets (Appendix A) and the floral diversity of *Dianthus* (Figure 1), illustrating that the genus *Dianthus* has undergone a recent and rapid evolutionary radiation, and the variation of molecular sequences has not been fully preserved, the length was short in most terminal nodes. Our findings were consistent with Greenberg et al. [39] and Valente et al. [40]. Similar studies have also been reported for *Lagerstroemia* [38], *Saussurea* [41], and Bambusoideae [42].

Highly variable regions can be used as potential DNA barcode markers for the studies on phylogenetic relationships, species identification, and population genetics [12,13,24,43,44,45,46]. Research has shown that all *Taxus* species can be successfully discriminated with 100% support using entire plastomes as super-barcodes, and *accD* and *rrn16*-*rrn23* were promising special barcodes to discriminate new species [47]. Moreover, the mini-barcode of primers ZJ818F-1038R (*ycf1b*) were proven to precisely discriminate between *Gleditsia sinensis* and *G. japonica* and reflect their biomass ratios accurately [43]. In the fern genus *Adiantum* from China, the two-barcode combination of *rbcL* + *trnH*-*psbA* was considered to be the best choice for barcoding *Adiantum* [48]. These results indicate that DNA markers developed based on chloroplast genomes have high resolution in distinguishing between genera or species. We compared the sequence divergences of the 12 newly assembled individuals in *Dianthus* species, with genes such as *psaA*, *ycf3*, *clpP*, *psbT*, *psbN*, *rpl16*, *rps19*, *ycf1*, and *ndhA* and tRNAs such as *trnT-GGU*, *trnT-UGU*, *trnL-UAA*, *trnV-UAC*, *trnM-CAU*, and *trnL-UAG,* showing a relatively high degree of sequence divergence. Additionally, several intergenic spacer regions were highly diverse, which were *rps16-trnR-UCU*, *aptF-aptI*, *rpoB-psbD*, *rps4-trnL-UAA*, *trnP-UGG-psbB*, *trnS-UGA-rps14*, *rpl16-rpl2*, and *rpl32-ycf1* (Appendix A). Furthermore, by calculating the Pi value, it is obvious that the *atpB* gene and *ycf1* gene had high Pi values, which indicate that the vicinities of the *atpB* gene and *ycf1* gene are probably hypervariable regions (Figure 7). Based on the above analysis, we selected several regions where sequences had a high degree of divergence to develop DNA markers. Additionally, we found that one DNA marker can be used to differentiate *D. caryophyllus* (Dca ‘XSZH’), *D. barbatus* (Dbr ‘XB’), and two cultivars (‘WC’ and ‘HY’) from *D. superbus* (Dsu ‘QM’), *D. chinensis* (Dch ‘MH’, Dch ‘DPD’, Dch ‘X’, Dch ‘L’, Dch ‘dhs’, and Dch ‘cf’), and hybrid offspring F_1_ (F_1_ ‘87M’) (Figure 8). This result was highly consistent with the results of the phylogenetic analysis, which verifies the correctness and rationality of the results of the phylogenetic analysis.

## 4. Materials and Methods

### 4.1. Plant Materials, DNA Extraction, and Sequencing

We collected 12 individuals from *Dianthus*, which included nine inbred lines cultivated by our lab for many years; one was the hybrid offspring F_1_ from interspecific hybridization, and the other two were cultivated species from online shop (Taobao, Xuzhou, Jiangsu Province, China) (Figure 1). Specific sample information is listed in Appendix A. These samples were in the flower cultivation base of Huazhong Agricultural University (located at 30°28′36.5″ N and 114°21′59.4″ E), Wuhan, Hubei Province, China.

The fresh leaves of samples were frozen at −80 °C before DNA extraction. Total genomic DNA was extracted via the phenol chloroform isoamyl alcohol extraction method. The quantified DNA of all individuals was used to construct Illumina libraries with average insert sizes of 350 bp and sequenced using the Illumina NovaSeq 6000 platform (Illumina, San Diego, CA, USA) in accordance with the manufacturer’s manual.

### 4.2. De Novo Genome Assembly and Annotation

Before assembly, low-quality data and adaptors were removed from the obtained Illumina paired-end total DNA sequencing data using Fastp (version 0.20.1, Department of Bioinformatics, HaploX Biotechnology, Shenzhen, China and Shenzhen Institutes of Advanced Technology, Chinese Academy of Sciences, Shenzhen, China) [49]. Then, the remaining high-quality reads were assembled into contigs using GetOrganelle software (version 1.7.1, Germplasm Bank of Wild Species, Kunming Institute of Botany, Chinese Academy of Sciences, Kunming, Yunnan, China) with k-mer 21, 45, 65, 85, 105 [50]. Except for the Dch ‘X’ sample, the rest of the samples can form a circular structure. The two gaps for the Dch ‘X’ sample were filled via Sanger sequencing, with specific primers designed for PCR (Polymerase chain reaction). All the primers used are listed in Appendix A. All the assembled chloroplast annotations were performed with the CPGAVAS2 online tool (http://47.96.249.172:16019/analyzer/annotate, (accessed on 17 February 2022)), using the default parameters to predict protein coding genes, tRNA genes, and rRNA genes [51]. Manual verification was performed using Apollo software (version 1.11.8, Environmental Genomics and Systems Biology, Lawrence Berkeley National Laboratory, California, USA). The boundaries of the introns and the start and stop codons were manually corrected. Annotations of tRNAs were confirmed using BLASTn (https://blast.ncbi.nlm.nih.gov/Blast.cgi, (accessed on 17 February 2022)). The annotated chloroplast genomes of *Dianthus* were submitted to GenBank (GenBank accession numbers: Dch ‘cf’, OP136016; F_1_‘87M’, OP136017; Dch ‘dhs’, OP136018; Dch ‘DPD’, OP136019; ‘HY’, OP136020; Dch ‘L’, OP136021; Dch ‘MH’, OP136022; Dsu ‘QM’, OP136023; ‘WC’, OP136024; Dch ‘X’, OP136025; Dbr ‘XB’, OP136026; and Dca ‘XSZH’, OP136027).

Phanta Max Super-Fidelity DNA polymerase (P505-d1, Vazyme, Nanjing, China) was used for PCR amplifications. For the amplicon PCR, the 50 μL reaction mix contained 2 mM Phanta Max Buffer, 10 µM of each primer, 100 ng template DNA, 10 µM dNTP mix, 1 μL Phanta Max Super-Fidelity DNA Polymerase, and 17 μL ddH_2_O. The PCR program was performed as follows: denaturation at 95 °C for 3 min, followed by 35 cycles at 95 °C for 15 s, the specific annealing temperature (Tm) for 15 s, 72 °C for 30 s, and 72 °C for 1 min as the final extension. PCR amplicons were visualized on 1.5% agarose gels and then sent to Sangon Biotech (Shanghai) for sequencing (Appendix A).

### 4.3. Repeated Sequence Analysis and Comparison of Genome Structures

The total lengths of the assembly chloroplast genomes and the GC contents were analyzed using the software seqkit (version 0.13.2, Department of Microbiology, College of Basic Medical Sciences, Third Military Medical University, 30#Gaotanyan St., Shapingba District, Chongqing, China) [52]. SSRs were identified using the online tool MISA (version 2.1, Leibniz Institute of Plant Genetics and Crop Plant Research Gatersleben, Seeland, Germany) (https://webblast.ipk-gatersleben.de/misa/index.php?action=1, (accessed on 18 February 2022)) [53,54]. Microsatellites were detected with thresholds of 10 repeat units for mono-, six repeat units for di-, four repeat units for tri- and tetra-, and three repeat units for penta- and hexanucleotide SSRs. Tandem repeats were analyzed using the Tandem repeats finder (version 4.09, Department of Biomathematical Sciences, Mount Sinai School of Medicine, New York, USA) (https://tandem.bu.edu/trf/trf.basic.submit.html, (accessed on 18 February 2022)) [55]. Dispersed repeats were detected with REPuter (https://bibiserv.cebitec.uni-bielefeld.de/reputer/, (accessed on 18 February 2022)) with a minimal length of 30 bp and a hamming distance of 3 [56]. Additionally, a comprehensive comparison at the LSC/IR/SSC boundaries was performed among the *Dianthus* species though IRscope (https://irscope.shinyapps.io/irapp/, (accessed on 4 May 2022)), which is a tool for visualizing the genes on the boundaries of the junction sites of the chloroplast genome [57].

### 4.4. Evolutionary Rates Analysis

To investigate the rates at which different plastid genes evolve, the Ks of 76 protein coding genes in 16 individuals in seven *Dianthus* species (Appendix A) was calculated seriatim under the branch model in PAML (version 4.9j, Department of Biology, Galton Laboratory, University College London, London, United Kingdom) [58,59]. First, the 76 protein coding genes of 16 individuals were extracted from their genomes. Then, each gene sequence of the 16 individual chloroplast genomes was aligned in MAFFT (version 7.313, Immunology Frontier Research Center, Osaka University, Suita, Osaka, Japan and Computational Biology Research Center, The National Institute of Advanced Industrial Science and Technology (AIST), Tokyo, Japan) [60]. Next, MEGA7 (version 7, Research Center for Genomics and Bioinformatics, Tokyo Metropolitan University, Hachioji, Tokyo, Japan and Department of Biological Sciences, Tokyo Metropolitan University, Hachioji, Tokyo, Japan) was used to construct an evolutionary tree of each gene with the parameters of the Kimura 2-parameter model and bootstrap replications of 1000 [61]. Finally, based on the branch model from MEGA7, the rates of Ka, Ks, and their ratio (Ka/Ks, denoted ω) for each gene of 16 *Dianthus* chloroplast genomes were calculated using the program CODEML from PAML [58,59].

### 4.5. Phylogenomic Analysis

For phylogenomic analysis, we constructed phylogenetic relationships, both in *Dianthus* and in other species in Caryophyllales, in order to obtain more detailed and accurate results. Targeting the *Dianthus* genus, *D. caryophyllus*, *D. gratianopolitanus*, *D. chinensis*, and *D. longicalyx* chloroplast genomes were downloaded from NCBI (https://www.ncbi.nlm.nih.gov/genome/browse#!/organelles/, (accessed on 30 September 2021)), and we selected *Psammosilene tunicoides* and *Gypsophila vaccaria*, which are close genera with *Dianthus*, as the outgroups (Appendix A). As for Caryophyllales, in addition to 16 individual chloroplast genomes (four had been reported before and 12 were reported in this study), a total of 30 plastome accessions were selected from NCBI, including one for each species of Amaranthaceae, Achatocarpaceae, Nepenthaceae, and Polygonaceae, and 23 Caryophyllaceae species, as well as another three Chenopodiaceae species as outgroups (Appendix A). We constructed two datasets; one only contained *Dianthus* species as well as *P. tunicoides* and *G. vaccaria*, and we refer to this as the *Dianthus* dataset in later studies. The other contained both *Dianthus* species and other species in Caryophyllales, and we refer to this as the Caryophyllales dataset in later studies. For the *Dianthus* dataset and the Caryophyllales dataset, three matrices with different strategies were constructed respectively, i.e., a) whole chloroplast genomes: all chloroplast genome sequences were aligned in MAFFT and trimmed using Gblocks (version 0.91b, Department of Physiology and Molecular Biodiversity, Institute of Molecular Biology of Barcelona, Barcelona, Spain) with default parameters; b) CDS concatenation: all shared CDS sequences were aligned in MAFFT and concatenated using PhyloSuite (version 1.2.2, Key Laboratory of Aquaculture Disease Control, Ministry of Agriculture, and State Key Laboratory of Freshwater Ecology and Biotechnology, Institute of Hydrobiology, Chinese Academy of Sciences, Wuhan, China); and c) exclude-third-codon-site matrix, with deletion of the terminal base of each codon [60,62,63].

For each matrix, ML analyses were implemented in IQ-TREE (version 1.6.8, Center for Integrative Bioinformatics Vienna, Max F. Perutz Laboratories, University of Vienna, Medical University of Vienna, Vienna, Austria) with 1000 replications for the bootstrap (BS) calculation, accounting for clade credibility [64]. BI analyses were performed in Mrbayes (version 3.2.7, Department of Biodiversity Informatics, Swedish Museum of Natural History, Stockholm, Sweden), with two independent runs consisting of one cold chain and three incrementally heated chains. Each run was conducted with 20,000 generations, sampling was performed every 100 generations, and a 25% burn-in was used [65]. The optimal model was determined through ModelFinder, using the built-in PhyloSuite program, and using the corrected Akaike Information Criterion [63,66]. The final trees were visualized and beautified on the iTOL v6 online site (https://itol.embl.de/, (accessed on 22 March 2022)) [67].

### 4.6. Identification of the Hypervariable Regions

We conducted a comparative genome analysis for the complete *Dianthus* plastomes, using the software Mvista (http://genome.lbl.gov/vista/mvista/submit.shtml, (accessed on 8 May 2022)) in the Shuffle-LAGAN mode [68,69]. The annotated Dca ‘XSZH’ plastome was used as the reference in the analysis. To identify the most divergent regions, MAFFT was used to align the complete *Dianthus* chloroplast genome sequences [60]. Next, we analyzed the DNA polymorphism using DnaSP (version 6.12.03, Departament de Genètica, Microbiologia i Estadıstica and Institut de Recerca de la Biodiversitat, Universitat de Barcelona, Barcelona, Spain), with a 500 bp window length and a 500 bp step size in haploid mode [70].

### 4.7. Identification and Validation of Molecular Markers for Species Discrimination

To discriminate among the 12 individuals in seven *Dianthus* species, we used polymorphisms in the hypervariable regions of chloroplasts to develop molecular markers. Specific primers were designed using the Primer3 program (http://bioinfo.ut.ee/primer3-0.4.0/, (accessed on 4 May 2022)) (Appendix A) [71,72]. In order to make the results more accurate and easier to distinguish, we used the method of capillary electrophoresis. The PCR amplifications and programs were the same as those described in the section of de novo genome assembly and annotation.

## 5. Conclusions

In this study, we reported the chloroplast genomes of 12 individuals in seven *Dianthus* species. Among them, *D. barbatus* (Dbr ‘XB’), *D. superbus* (Dsu ‘QM’), two cultivars (‘HY’, ‘WC’), and one hybrid (F_1_ ‘87M’) are reported for the first time. Through genome comparative analysis, we obtained the characteristic sequence regularity of the *Dianthus* chloroplast genome. At the same time, we clarified the evolutionary relationships between 16 individuals in nine *Dianthus* species by constructing the phylogenetic trees of *Dianthus* and Caryophyllales. The phylogenetic analysis supported 16 individuals and produced two sister clades (Clade A and Clade B) with very high support. Clade A contained five species, namely *D. caryophyllus*, *D. barbatus*, *D. gratianopolitanus*, and two cultivars (‘HY’ and ‘WC’). Clade Bincluded four species, in which *D. superbus* was a sister branch with *D. chinensis*, *D. longicalyx*, and F_1_ ‘87M’ (the hybrid offspring F_1_ from *D. chinensis* and ‘HY’). Furthermore, based on an analysis of the hypervariable regions, one DNA marker was developed to identify the two major clades of the 16 individuals in nine *Dianthus* species in this study. Taken together, our results provide additional information for our understanding of *Dianthus* classification and chloroplast genome evolution.

## Figures and Tables

**Figure 1 ijms-23-12567-f001:**
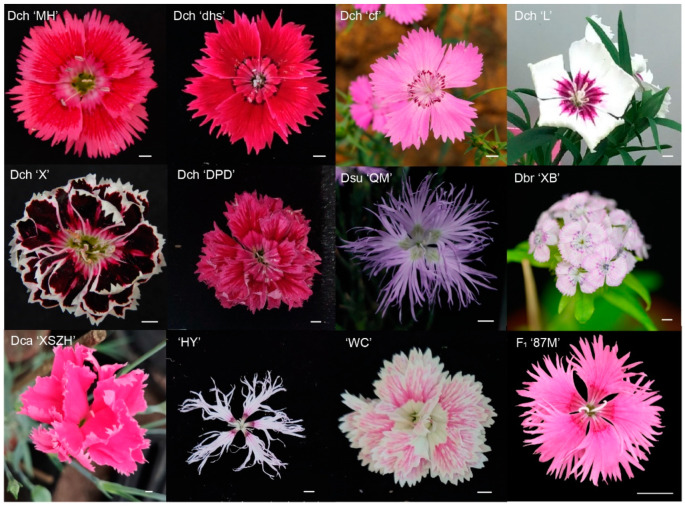
Morphological characteristics of flowers of *Dianthus* species in this study. Scale bar = 1 cm.

**Figure 2 ijms-23-12567-f002:**
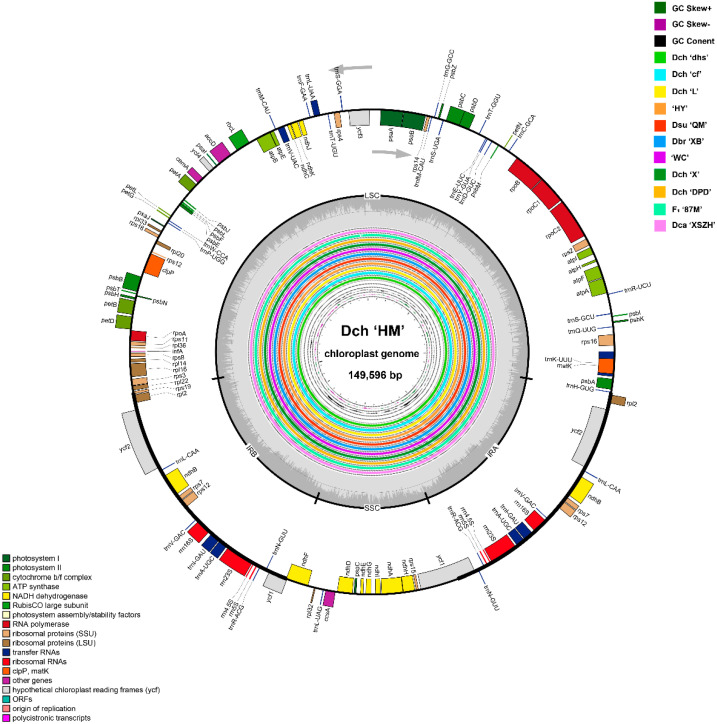
Chloroplast genome map of Dch ‘MH’ (the outermost three rings) and CGView [15] comparison of eleven *Dianthus* chloroplast genomes (the inter rings with different colors). Genes belonging to different functional groups are shown in different colors in the outermost first ring. Genes shown on the outside of the outermost first ring are transcribed counter-clockwise, and on the inside, clockwise. The gray arrowheads indicate the direction of the genes. The tRNA genes are indicated by one letter code of amino acids with anticodons. LSC, large single copy region; IR, inverted repeat; SSC, small single copy region. The innermost first black ring indicates the chloroplast genome size of Dch ‘MH’. The innermost second and third rings indicate GC skews and GC content deviations in the chloroplast genome of ‘MH’, respectively: GC Skew+ indicates G > C, and GC Skew– indicates G < C. From the innermost forth color ring to outwards 14th ring in turn: Dch ‘dhs’, Dch ‘cf’, Dch ‘L’, ‘HY’, Dsu ‘QM’, Dbr ‘XB’, ‘WC’, Dch ‘X’, Dch ‘DPD’, F_1_ ‘87M’, Dch ‘L’, and Dch ‘XSZH’; chloroplast genome similar and highly divergent locations are represented by continuous and interrupted track lines, respectively.

**Figure 3 ijms-23-12567-f003:**
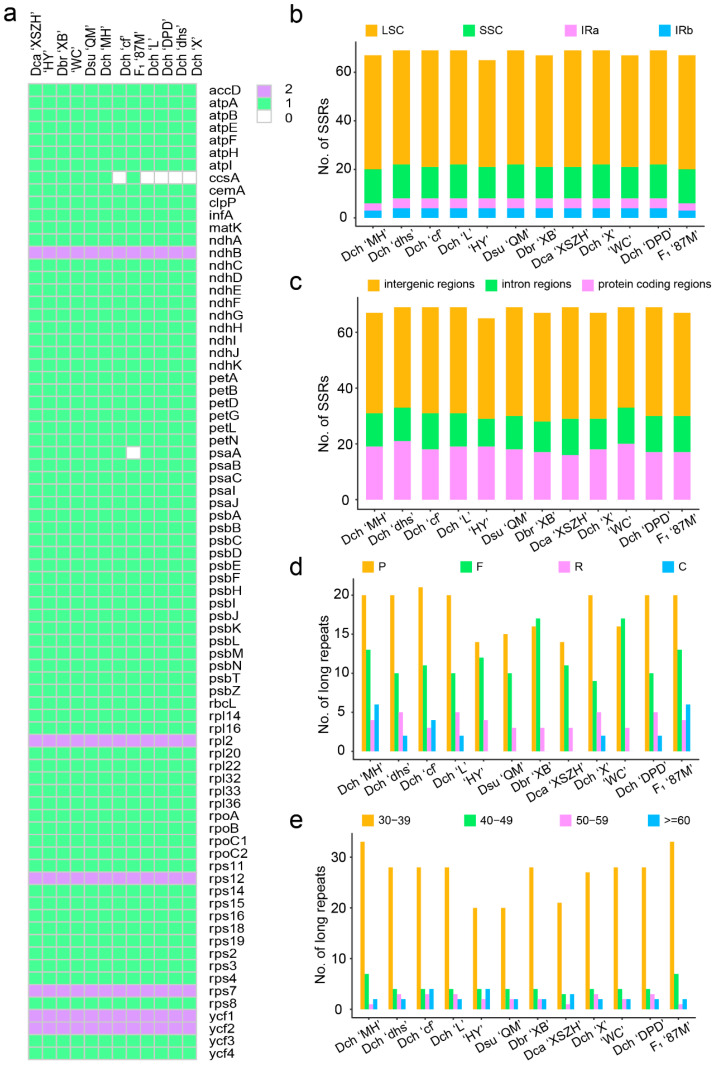
Protein coding gene content and repetitive sequence feature of 12 *Dianthus* chloroplast genomes. (**a**) Protein coding gene content of 12 *Dianthus* chloroplast genomes. Different colors represent the number of genes appearing in each chloroplast genome. (**b**) The frequencies of the identified SSRs in the LSC/SSC/IR regions. (**c**) The SSR distribution in protein coding regions, intron regions, and intergenic regions detected in 12 *Dianthus* chloroplast genomes. (**d**) A total of four long repeat types in 12 *Dianthus* chloroplast genomes. F, Forward; R, Reverse; C, Complement; P, Palindromic. (**e**) The numbers of long repeat sequences by length.

**Figure 4 ijms-23-12567-f004:**
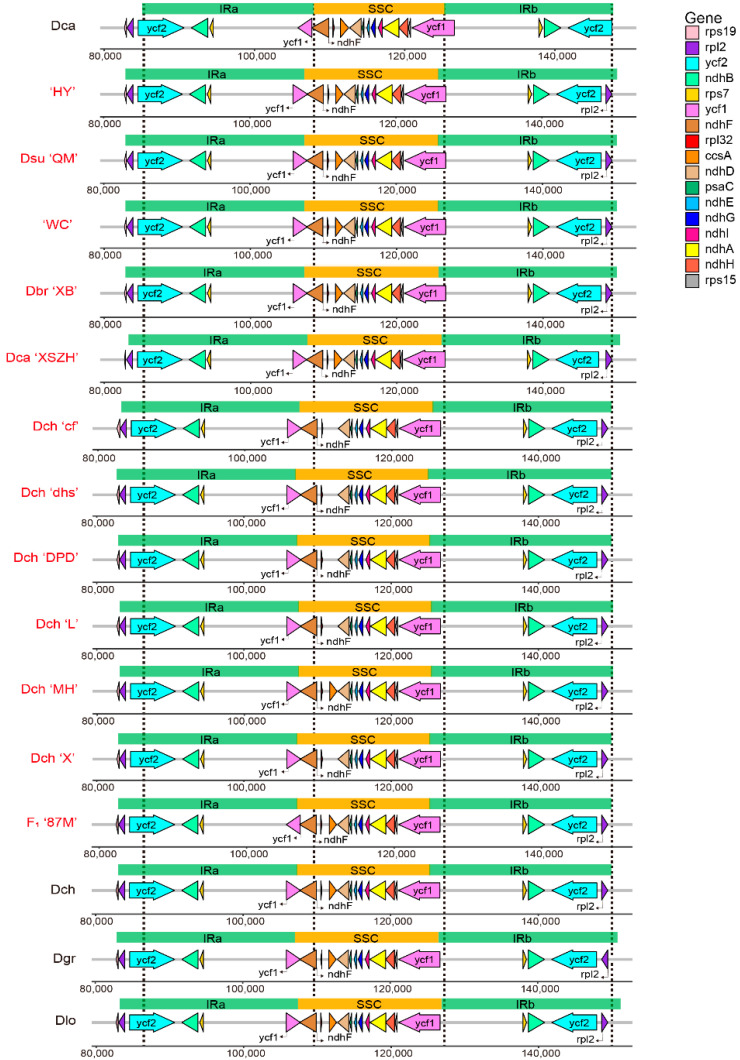
Comparisons of IR regions boundaries among 16 chloroplast genomes in *Dianthus*. The four dotted lines in the figure represent the boundary position of Dca. The 12 sequenced chloroplast genomes in this study are marked in red.

**Figure 5 ijms-23-12567-f005:**
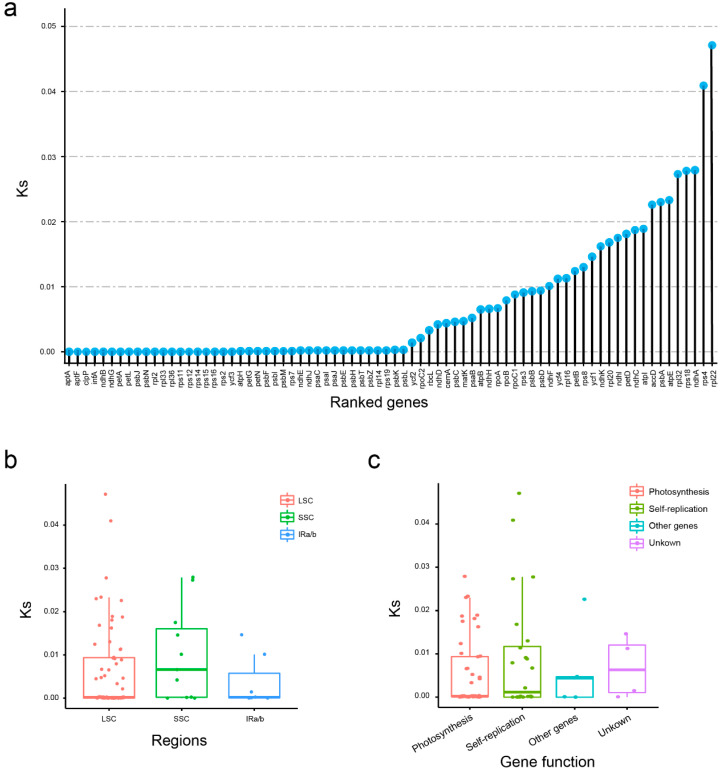
Compositions of Ks values of 76 protein coding genes in 12 *Dianthus* chloroplast genomes. (**a**) Ks values of 76 protein coding genes ranked by Ks. (**b**) Ks values in LSC/SSC/IR regions. (**c**) Ks values in genes with different functional classification.

**Figure 7 ijms-23-12567-f007:**
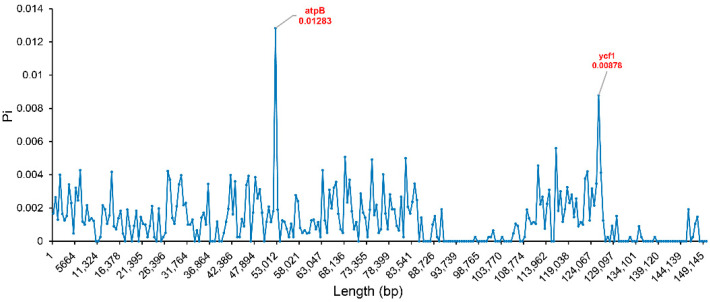
Sliding window analysis of the 16 *Dianthus* chloroplast genomes (step size, 500 bp; window length, 500 bp). x-axis: midpoint position of the window; y-axis: nucleotide diversity in each window.

**Figure 8 ijms-23-12567-f008:**
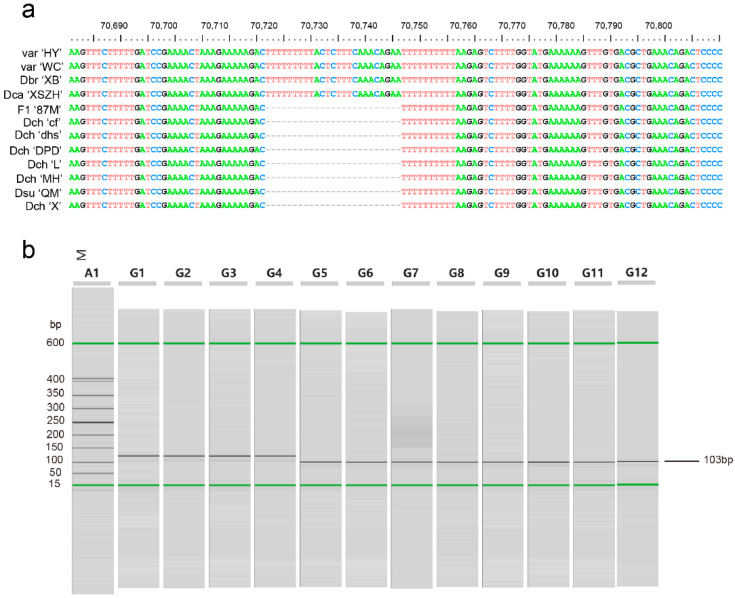
The capillary electrophoresis results of the amplification of DNA makers using designed primers. (**a**) Sequence alignment results at the positions of the design Maker1. (**b**) The result of Maker1. Lane M is the marker of DL400. The lanes from left to right correspond to products amplified from Dca ‘XSZH’, ‘HY’, Dbr ‘XB’, ‘WC’, Dsu ‘QM’, Dch ‘MH’, F_1_ ‘87M’, Dch ‘cf’, Dch ‘dhs’, Dch ‘X’, Dch ‘L’, and Dch ‘DPD’.

**Table 1 ijms-23-12567-t001:** Characteristics of the *Dianthus* chloroplasts generated in this study.

Sample	Genome Size /bp	GC Content /%	LSC Length /bp	SSC Length /bp	IR Length /bp	Gene (Protein, rRNA, tRNA)
Dch ‘MH’	149,596	36.33	82,840	17,150	24,803	126 (84, 8, 34)
Dch ‘dhs’	149,641	36.31	82,886	17,139	24,808	125 (83, 8, 34)
Dch ‘cf’	149,602	36.32	82,837	17,149	24,808	125 (83, 8, 34)
Dch ‘L’	149,641	36.31	82,886	17,139	24,808	125 (83, 8, 34)
Dch ‘X’	149,641	36.31	82,883	17,139	24,808	125 (83, 8, 34)
Dch ‘DPD’	149,641	36.31	82,888	17,139	24,807	125 (83, 8, 34)
Dsu ‘QM’	149,726	36.30	82,954	17,135	24,818	126 (84, 8, 34)
Dbr ‘XB’	149,642	36.31	82,893	17,123	24,813	124 (84, 8, 33)
Dca ‘XSZH’	149,596	36.32	82,935	17,096	24,781	124 (84, 6, 34)
‘WC’	149,660	36.31	82,909	17,123	24,814	126 (84, 8, 34)
‘HY’	149,800	36.30	82,963	17,227	24,805	126 (84, 8, 34)
F_1_ ‘87M’	149,192	36.30	82,436	17,150	24,803	125 (83, 8, 34)

**Table 2 ijms-23-12567-t002:** Genes present in the 12 sequenced chloroplast genomes in *Dianthus*.

Category of Genes	Group of Genes	Name of Genes
Photosynthesis	Subunits of ATP synthase	atpA, atpB, atpE, atpF *, atpI, atpH
	Subunits of photosystem II	psbA, psbB, psbC, psbD, psbE, psbF, psbI, psbH, psbJ, psbK, psbL, psbM, psbN, psbT, psbZ
	Subunits of NADH-dehydrogenase	ndhA *, ndhB (×2) *, ndhC, ndhD, ndhE, ndhF, ndhG, ndhH, ndhI, ndhJ, ndhK
	Subunits of cytochrome b/f complex	petA, petB *, petD *, petG, petL, petN
	Subunits of photosystem I	psaA ②, psaB, psaC, psaI, psaJ, ycf3
	Subunits of rubisco	rbcL
Self-replication	Large subunit of ribosome	rpl2 (×2), rpl14, rpl16 *, rpl20, rpl22, rpl32, rpl33, rpl36
	DNA dependent RNA polymerase	rpoA, rpoB, rpoC1 *, rpoC2
	Small subunit of ribosome	rps2, rps3, rps4, rps7 (×2), rps8, rps11, rps12 (×2) **, rps14, rps15, rps16 *, rps18, rps19 ④
	Ribosomal RNAs	rrn4.5 (×2), rrn5 (×2), rrn16 (×2), rrn23 (×2)
	Transfer RNAs	trnA-UGC (×2) *, trnC-GCA, trnD-GUC, trnE-UUC, trnF-GAA, trnfM-CAU, trnG-GCC, trnH-GUG ⑤, trnI-GAU (×2) *, trnK-UUU*, trnL-CAA (×2), trnL-UAA *, trnL-UAG, trnM-CAU, trnN-GUU (×2), trnP-UGG, trnQ-UUG, trnR-ACG (×2), trnR-UCU, trnS-GCU, trnS-GGA, trnS-UGA, trnT-GGU, trnT-UGU, trnV-GAC (×2), trnV-UAC *, trnW-CCA, trnY-GUA
Other genes	Subunit of acetyl-CoA-carboxylase	accD
	c-type cytocgrom synthesis gene	ccsA ①
	Envelop membrane protein	cemA
	Translational initiation factor	infA
	Protease	clpP **
	Maturase	matK
Unkown	Conserved open reading frames	ycf1 (×2) ③, ycf2 (×2), ycf3 **, ycf4

Note: (×2): gene with two copies; *: gene containing one intron; **: gene containing two introns; ①: *ccsA* gene is missing in the chloroplast genomes of Dch ‘cf’, Dch ‘L’, Dch ‘DPD’, Dch ‘dhs’ and Dch ‘X’; ②: *psaA* gene is missing in the chloroplast genomes of F_1_ ‘87M’; ③: *ycf1* had one intron in the chloroplast genome of Dca ‘XSZH’ and ‘HY’ and no intron in the other 10 sequenced chloroplast genomes in this study; ④: *rps19* had one intron in the chloroplast genome of Dch ‘cf’, with the protein length of *rps19* in Dca ‘XSZH’ of only 49, and no intron in the other 10 sequenced chloroplast genomes in this study; ⑤: *trnH-GUG* is missing in the chloroplast genomes of Dbr ‘XB’.

**Table 3 ijms-23-12567-t003:** Type and number of SSRs found in the *Dianthus* chloroplasts.

Type	Repeat Unit	Numbers of Repeats											
		Dch ‘MH’	Dch ‘dhs’	Dch ‘cf’	Dch ‘L’	Dch ‘X’	Dch ‘DPD’	Dsu ‘QM’	Dbr ‘XB’	Dca ‘XSZH’	‘HY’	‘WC’	F_1_ ‘87M’
Mono-	A/T	62	64	64	64	64	64	63	62	66	61	62	64
	C/G	1	2	1	2	2	2	1	1	1	1	1	1
Di-	AT/AT	3	3	3	3	3	3	2	3	2	2	3	3
Tri-	AAT/ATT	9	7	6	7	7	7	8	8	8	7	8	9
Tetra-	AAAT/ATTT	1	1	1	1	1	1	1	0	0	0	0	1
Penta-	AACAC/GTGTT	0	2	2	2	2	2	2	2	2	2	2	0
	AATAC/ATTGT	1	1	0	1	1	1	1	1	1	1	1	1
Hexa-	AAATAT/ATATTT	0	0	1	0	0	0	0	0	0	0	0	0
	AATATG/ATATTC	0	0	0	0	0	0	0	1	0	1	1	0
	AATATT/AATATT	0	0	0	0	0	0	1	0	0	0	0	0
Total No.		77	80	78	80	80	80	79	78	80	75	78	79

## Data Availability

Not applicable.

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
