# Peer review of "Comprehensive Comparative Analysis and Development of Molecular Markers for Dianthus Species Based on Complete Chloroplast Genome Sequences"

_ijms, 2022, doi:10.3390/ijms232012567_

Round 1
Reviewer 1 Report
The authors are presenting the Manuscript entitled "Comprehensive comparative analysis and development of mo-lecular markers for Dianthus species based on complete chlo-roplast genome sequences". The Manuscript is well written, the methods exhaustive and the results well presented.
I just have few minor comments, listed below.
- Remove "and so on" on line 50, it's not that professional.
- Refrase the sentences regarding the use of Dianthus species. It sounds like a repetition (line 41 with lines 51 ...)
- Maybe it's better to move all the chloroplast genome sequence references and evidence to the Discussion (Lines 66 - 98). You could just introduce some past application but better discuss later in the Discussion.
- Be careful with the missing spaces before citation (see lines 75 before [9] or line 90 before [13]
- line 468 - 474 : For PCR reproducibility, please use concentration instead of volumes used for PCR reactions, in a total final volume.
- line 475, the title paragraph is a bit misleading since the paragraph is also describing the repeats analysis.
Reviewer 2 Report
Authors have written the manuscript well and have provided all the necessary files for verification. The only major concern is that the GenBank accession IDs of the 12 chloroplast genome provided in their study could not be verified. Hence authors are advised to check them again and provide an appropriate explanation. Other minor corrections are suggested below:
Line 90: new is misspelled. Please check.
Figure 1 is missing.
Fig 3a shows more than 12 individuals so please check
In Fig 6 legend, please check the sentence on line 274 & 275. Please replace “were used” with “used”.
Please remove the brackets in line 265 and 269
In Figure 8, replace “amplificated” with amplified.
Sentence in line 413 seems incomplete, please check and rephrase it.
In line 418, it is radiation?
Reviewer 3 Report
In the manuscript entitled “Comprehensive comparative analysis and development of molecular markers for Dianthus species based on complete chloroplast genome sequences" the main objective of the authors was to compare different Dinathus species based on their chloroplast genome sequences. The study was well planned and performed. However, some minor corrections should be considered as outlined below before publication of the manuscript.
Abstract: Authors could more focus on their findings instead of explaining the plant importance.
Line 14: use economic and ornamental value instead of “economic value and ornamental value”.
Line 25-27: rewrite the sentence.
Line 40: remove or replace the word “good”
Line 47: cite an appropriate reference.
Line 126: figure 2? How about the figure 1? Rearrange figure orders and numbers.
Line 184: “F1” use subscript format of the number. Apply this throughout the manuscript.
Line 154: If the citation format is based on the number order of the reference which appears in the manuscript, authors
Conclusion: In this section author are advised to show more major findings of their study.
